# Association of Tumor PD-L1 Expression with the T790M Mutation and Progression-Free Survival in Patients with EGFR-Mutant Non-Small Cell Lung Cancer Receiving EGFR-TKI Therapy

**DOI:** 10.3390/diagnostics10121006

**Published:** 2020-11-25

**Authors:** Minehiko Inomata, Kenji Azechi, Naoki Takata, Kana Hayashi, Kotaro Tokui, Chihiro Taka, Seisuke Okazawa, Kenta Kambara, Shingo Imanishi, Toshiro Miwa, Ryuji Hayashi, Shoko Matsui, Kazuyuki Tobe

**Affiliations:** 1First Department of Internal Medicine, Toyama University Hospital, Toyama 930-0194, Japan; cont2dct@gmail.com (K.A.); leftsideback2@gmail.com (N.T.); canape@med.u-toyama.ac.jp (K.H.); tokuiktr@med.u-toyama.ac.jp (K.T.); mameciroro@gmail.com (C.T.); okazawas@med.u-toyama.ac.jp (S.O.); kkent@med.u-toyama.ac.jp (K.K.); shingo@med.u-toyama.ac.jp (S.I.); mtoshi-tym@umin.ac.jp (T.M.); smatsui@med.u-toyama.ac.jp (S.M.); tobe@med.u-toyama.ac.jp (K.T.); 2Department of Medical Oncology, Toyama University Hospital, Toyama 930-0194, Japan; hsayaka@med.u-toyama.ac.jp

**Keywords:** epidermal growth factor receptor, lung cancer, T790M, prognosis, programmed death ligand-1

## Abstract

Background: Among patients with non-small cell lung cancer (NSCLC), we compared the progression-free survival (PFS) and proportion of acquisition of T790M mutation of the epidermal growth receptor gene (EGFR) after first-line treatment with epidermal growth factor receptor-tyrosine kinase inhibitors (EGFR-TKIs) in patient groups with and without tumor expression of programmed death ligand-1 (PD-L1). Methods: Data of patients with EGFR-mutant NSCLC were retrospectively analyzed. Tumor PD-L1 expression was evaluated by immunohistochemistry using the 22C3 antibody. T790M gene mutation was evaluated by Cobas EGFR assay using tissues or humoral specimens. Results: Data of 47 patients with EGFR-mutant NSCLC were analyzed. The median (95% confidence interval) PFS in the PD-L1-negative and -positive patient groups were 12.9 (9.7–15.4) months and 9.0 (5.1–12.3) months, respectively (*p* = 0.029). T790M gene mutation was analyzed in 27 patients. The proportion of acquisition of T790M mutation of EGFR after first-line treatment with an EGFR-TKI was higher in the PD-L1-negative patient group than in the PD-L1-positive patient group (8/11 patients (72.7%) vs. 4/16 patients (25.0%); *p* = 0.022). Conclusions: Patients with negative tumor PD-L1 expression showed longer PFS and a higher proportion of acquisition of T790M mutation of EGFR after first-line treatment with an EGFR-TKI.

## 1. Introduction

In patients with EGFR-mutant non-small cell lung cancer (NSCLC), treatment with epidermal growth factor receptor-tyrosine kinase inhibitors (EGFR-TKIs) has been shown to yield significantly longer progression-free survival (PFS) periods as compared to treatment with cytotoxic agents [1,2]. However, most patients receiving EGFR-TKI treatment develop acquired resistance to the drug sooner or later. The most common mechanism of resistance is acquisition of the T790M mutation of EGFR; a phase III trial has demonstrated that treatment with osimertinib, a third-generation EGFR-TKI, was associated with significantly longer PFS, as compared to cytotoxic agents, in patients with acquired resistance to first-line EGFR-TKI treatment who had acquired T790M [3]. Previous studies have reported an association of the PFS with the presence/absence of T790M, and also associations of the duration of first-line EGFR-TKI treatment [4,5,6,7], and presence of exon 19 deletion of EGFR [5,7] with the likelihood of detection of T790M.

Tumor expression of programmed death ligand-1 (PD-L1) has been shown to be associated with the efficacy of immune checkpoint inhibitors in patients with non-squamous NSCLC [8], whereas immune checkpoint inhibitors are known to be less effective in patients with EGFR-mutant NSCLC [9,10]. On the other hand, there are several reports on the relationship between tumor PD-L1 expression and the PFS after EGFR-TKI treatment [11,12,13,14,15]. Although inconsistent results have been reported, tumor PD-L1 expression could affect the clinical course in patients with EGFR-mutant NSCLC. Furthermore, EGFR-mutant NSCLC patients who developed resistance to EGFR-TKIs via acquisition of T790M showed lower tumor PD-L1 expression levels and also a lower proportion of patients with positive tumor PD-L1 expression [16,17].

Based on these previous reports, we analyzed the data of EGFR-mutant NSCLC patients to compare the PFS and proportion of acquisition of T790M after first-line treatment with EGFR-TKIs in patient groups with and without PD-L1 expression.

## 2. Materials and Methods

### 2.1. Study Design and Patient Selection

We reviewed the medical charts of patients with NSCLC treated at Toyama University Hospital and conducted a retrospective analysis of the data. The inclusion criteria were as follows: (1) patients with cytologically or histologically confirmed NSCLC harboring sensitive EGFR mutations; and (2) patients treated with EGFR-TKIs including gefitinib, erlotinib, or osimertinib as first-line treatment between 2014 and 2019. The exclusion criterion was patients whose records lacked data on the tumor PD-L1 status.

The PFS of patients with EGFR-mutant NSCLC treated with EGFR-TKIs was determined. The disease stage was evaluated according to the 8th edition of the UICC TNM classification, and patients who had undergone surgery were re-classified based on the findings at the start of the EGFR-TKI treatment. Then, analysis of the proportion of acquisition of T790M was conducted in patients in whom the T790M status was evaluated after failure of first-line treatment with a 1st- or 2nd-generation EGFR-TKI.

This study was conducted in accordance with the Declaration of Helsinki and the Ethical Guidelines for Medical and Health Research Involving Human Subjects (Ministry of Health, Labour and Welfare of Japan), after obtaining the approval of the Ethics committee, University of Toyama (Reference number: R2019180, 10 March 2020). The informed consent of the study was waived by the ethics committee of university of Toyama, and we disclosed information about the study instead of obtaining written informed consent.

### 2.2. Immunohistochemistry

PD-L1 expression was evaluated in formalin-fixed and paraffin-embedded tumor tissue specimens by immunohistochemistry using the 22C3 antibody (BML, Tokyo, Japan). The proportion of PD-L1-positive tumor cells was calculated as the tumor proportion score (TPS), and a TPS of 1% or more was defined as positive.

### 2.3. Polymerase Chain Reaction

The presence of T790M was evaluated by the Cobas EGFR real-time PCR assay (LSI Medience, Tokyo, Japan) using tissues or humoral specimens obtained after the patients developed resistance to the EGFR-TKI used (15 patients: tissue specimens, nine patients: blood samples, two patients: cerebrospinal fluid, and one patient: pleural effusion). The minimum detection sensitivity for T790M mutation is reported to be 0.1% when using a plasma sample (https://www.info.pmda.go.jp/downfiles/ivd/PDF/700025_22800EZX00011000_A_01_10.pdf).

### 2.4. Statistical Analysis

Patients were dichotomized according to categorical variables or the median levels. Survival curves were drawn by the Kaplan-Meier method.

PFS was calculated from the date of initiation of treatment with an EGFR-TKI until the date of diagnosis of disease progression or occurrence of death due to any cause, and censored at the last visit at which no disease progression was observed. The relationships between the PFS and clinical parameters were analyzed by the log-rank test. Multivariate analysis was performed using a Cox proportional hazard model. EGFR status, performance status, lactate dehydrogenase level, disease stage, and PD-L1 expression were planned to be used as independent variables. The associations of the clinical parameters with the proportion of acquisition of T790M after first-line treatment with an EGFR-TKI were analyzed by Fisher’s exact test. Statistical analysis was performed using JMP ver. 14.0.2 (SAS, Cary, NC, USA).

## 3. Results

### 3.1. Patient Selection

Between 2014 and 2019, 67 patients with EGFR-mutant NSCLC received first-line treatment with EGFR-TKIs. Of these, 20 cases whose records lacked the data on tumor PD-L1 expression were excluded, and the data of the remaining 47 patients were analyzed to determine the PFS. Then, after a total of 20 patients, including 12 patients who were treated with osimertinib, 18 patients whose records lacked data on the T790M gene mutation status, and 13 patients who did not show disease progression were excluded, and the data of the remaining 27 patients were analyzed to determine the presence/absence of T790M (Figure 1).

### 3.2. PFS

Table 1 shows the patient characteristics. Forty-four of the 47 patients (93.6%) had lung adenocarcinoma, two (4.3%) had NSCLC (not otherwise specified), and one (2.1%) had adenosquamous cell lung cancer. Exon 19 deletion was detected in 16 (34.0%) patients, exon 21 L858R in 28 (59.6%) patients, exon 18 G719A in two (4.3%) patients, and exon 21 L861Q in one (2.1%) patient. The TPS of PD-L1 was ≥1% in 29 of the 47 (61.7%) patients. Tumor PD-L1 expression was evaluated using tumor specimen obtained after the first-line treatment with an EGFR-TKI in five patients, because the tissue specimens obtained before the first-line treatment were not available.

Figure 2 shows the association between the tumor PD-L1 expression status and the PFS. The median (95% confidence interval) PFS periods in the PD-L1-negative and PD-L1-positive patients were 12.9 (9.7–15.4) months and 9.0 (5.1–12.3) months, respectively (*p* = 0.029, log-rank test). Analysis limited to patients in whom tumor PD-L1 expression was evaluated in tissue specimens obtained before the start of the first-line EGFR-TKI treatment showed similar results (median PFS, 12.9 vs. 7.4 months, *p* = 0.027, log-rank test). No significant associations of other baseline characteristics with the PFS were observed (Table 2). Multivariate analysis revealed a significant association between PD-L1 expression and PFS (Table 3).

PD-L1 status (positive vs. negative) was not associated with clinical parameters, including age (<70 vs. ≥70, *p* = 1.000), sex (male vs. female, *p* = 1.000), smoking history (yes vs. no, *p* = 0.211), histology (adenocarcinoma vs. others, *p* = 0.276), EGFR mutation (Exon 19 del vs. Exon 21 L858R vs. others, *p* = 0.789), PS (0–1 vs. ≥2, *p* = 0.716), LDH (<200 IU/L vs. ≥200 IU/L, *p* = 0.130), brain metastasis (yes vs. no, *p* = 0.759), and stage (IIIA-C/IVA vs. IVB, *p* = 1.000) (Fisher’s exact test).

In the present study, three patients with uncommon mutation were enrolled in the analysis, including two patients with exon 18 G719A and one patient with exon 21 L861Q. The log-rank test revealed that median PFS after the initiation of treatment with EGFR-TKI was as short as 7.7 months compared to patients with common sensitive EGFR mutation (*p* = 0.308, log-rank test). However, the Cox regression hazard model suggested that there is an association between tumor PD-L1 expression and PFS after the initiation of treatment with EGFR-TKI independently of EGFR mutation status.

### 3.3. Acquisition of T790M

Table 4 shows the relationships between the patient characteristics and the proportion of acquisition of T790M after patients had developed resistance to the first-line EGFR-TKI treatment. The presence/absence of T790M was evaluated using tissue specimens in 15 cases (55.6%) and humoral specimens in 12 cases (44.4%); T790M was detected in 12 patients (44.4%). The PD-L1-negative group showed a higher proportion of acquisition of T790M than the PD-L1-positive group (8/11 patients (72.7%) vs. 4/16 patients (25.0%); *p* = 0.022, Fisher’s exact test). Analysis limited to patients in whom the tumor PD-L1 expression was evaluated in tissue specimens obtained before the start of the first-line EGFR-TKI treatment showed similar results (7/10 patients (70.0%) vs. 3/12 patients (25.0%); *p* = 0.084, Fisher’s exact test). There were no significant associations between the patient characteristics, except for the tumor PD-L1 expression status, and the proportion of acquisition of T790M.

Figure 3 shows the associations of the EGFR gene mutation and tumor PD-L1 expression status with the proportion of acquisition of T790M. The PD-L1-negative patient group, both among patients with exon 19 del and exon 21 L858 mutations of EGFR, showed a higher proportion of acquisition of T790M (Exon 19 del: PD-L1 negative: 4/4 (100%) vs. PD-L1 positive: 2/5 (40.0%); exon 21 L858R; PD-L1-negative: 4/7 (57.1%) vs. PD-L1-positive: 2/10 (20.0%), *p* = 0.045, Fisher’s exact test).

## 4. Discussion

The results of the present study showed that negative tumor PD-L1 expression in patients with EGFR-mutant NSCLC as evaluated using the 22C3 antibody was associated with a longer PFS and a higher proportion of acquisition of T790M after first-line treatment with an EGFR-TKI in patients with EGFR-mutant NSCLC.

Several studies have evaluated the relationship between tumor PD-L1 expression and the efficacy of EGFR-TKI therapy in patients with EGFR-mutant NSCLC, but no consistent results have been reported. The present study, conducted using the 22C3 antibody for immunohistochemical evaluation of tumor PD-L1 expression, showed that positive tumor PD-L1 expression was associated with shorter PFS, consistent with the results reported by Soo et al., who used the SP142 antibody for evaluating tumor PD-L1 expression [14]. Lin et al. and D’Incecco et al. reported contrasting results; they found that the PFS was longer in patients with positive tumor PD-L1 expression. They used a common yet different antibody for the PD-L1 immunohistochemistry [12,13]. It is possible that the results of PD-L1 immunohistochemistry obtained using different antibodies reflect different biological aspects of the tumor.

It is reported that patients with acquired T790M after failure of treatment with an EGFR-TKI show lower tumor PD-L1 expression levels or a lower proportion of patients with positive tumor PD-L1 expression [16,17]. The findings of the present study seem to be in line with these previous reports and suggest that the tumor PD-L1 status could be used to predict the likelihood of acquisition of T790M. However, in five of our 27 patients in whom we evaluated the tumor PD-L1 expression, the evaluation was conducted in tumor tissue specimens obtained after the initiation of first-line EGFR-TKI treatment. Nevertheless, we obtained similar results when the analysis was limited to patients in whom the tumor PD-L1 status was evaluated in tumor tissue specimens obtained before the initiation of EGFR-TKI treatment.

Several studies have reported that the proportion of acquisition of T790M is higher in patients with exon 19 deletion mutation of EGFR [5,7]. The findings of our study (Figure 3) suggest that the tumor PD-L1 status is associated with the likelihood of acquisition of T790M, independent of the EGFR mutation, that is, exon 19 deletion or exon 21 L858R. However, because the sample size in the present study was small, it is necessary to verify the findings in a larger clinical study.

There were several limitations of the present study. First, the study was a retrospective study conducted in a small number of patients. Therefore, the statistical power was limited, and random errors and selection bias could have influenced the results of the analysis. Second, in five of the 27 patients, the tumor PD-L1 expression was evaluated using tissue samples obtained after the initiation of EGFR-TKI treatment, and there are reports to suggest that the tumor PD-L1 status could change during treatment [17,18]. Because of these limitations, verification of our results in a larger patient series is necessary.

In conclusion, in patients with EGFR-mutant NSCLC enrolled in this study, the presence/absence of tumor PD-L1 expression was associated with the PFS and the proportion of acquisition of the T790M mutation of EGFR after first-line treatment with EGFR-TKIs. Although the results must be verified further, evaluation of tumor PD-L1 expression as a biomarker in patients with EGFR-mutant NSCLC is expected in the future.

## Figures and Tables

**Figure 1 diagnostics-10-01006-f001:**
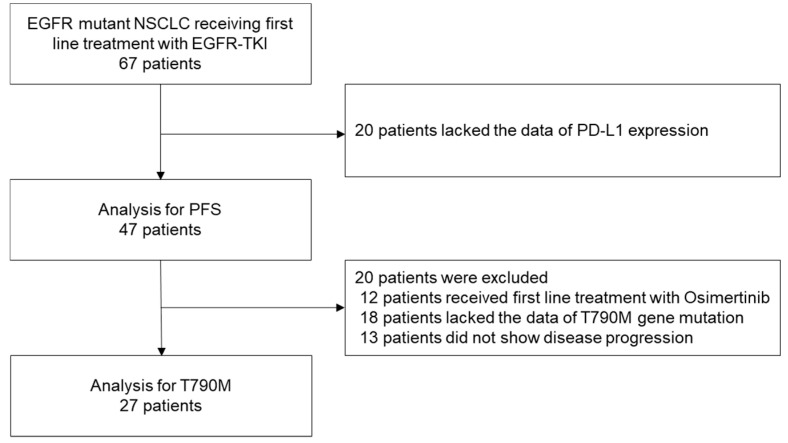
Patient selection. Of the data of 67 patients with EGFR-mutant NSCLC who had received first-line treatment with EGFR-TKIs, those of 47 patients were analyzed to determine the PFS and those of 27 patients were analyzed to determine the proportion of acquisition of the T790M mutation of EGFR.

**Figure 2 diagnostics-10-01006-f002:**
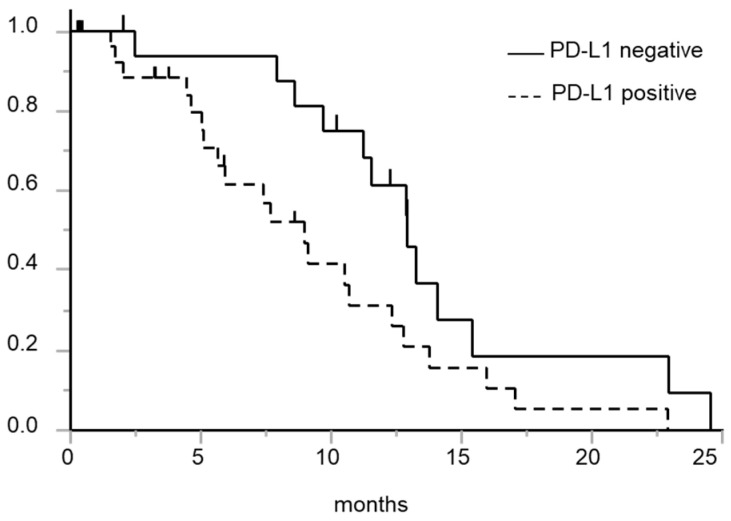
Kaplan-Meier curve for PFS after the initiation of first-line EGFR-TKI treatment in patients with and without tumor PD-L1 expression. Solid line: Patients with negative tumor PD-L1 expression; dashed line: Patients with positive tumor PD-L1 expression. The PD-L1-negative patient group showed a longer PFS than the PD-L1-positive patient group (*p* = 0.029, log-rank test).

**Figure 3 diagnostics-10-01006-f003:**
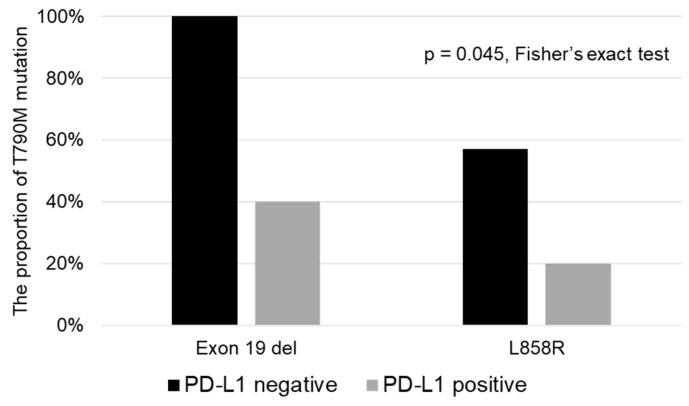
Associations of the EGFR gene mutation and tumor PD-L1 status with the proportion of acquisition of the T790M mutation of *EGFR.* In both patients with exon 19 del and exon 21 L858 mutations of *EGFR*, the PD-L1-negative patient subgroup showed a higher proportion of acquisition of the T790M mutation.

**Table 1 diagnostics-10-01006-t001:** Patient characteristics.

		47 (100%)
Age (yr)	<70	23 (48.9%)
	≥70	24 (51.1%)
Sex	Male	20 (42.6%)
	Female	27 (57.4%)
Smoking history	Yes	17 (36.2%)
	No	30 (63.8%)
PS	0–1	38 (80.9%)
	≥2	9 (19.1%)
Histology	Adenocarcinoma	44 (93.6%)
	Others	3 (6.4%)
EGFR mutation	Exon 19 Deletion	16 (34.0%)
	Exon 21 L858R	28 (59.6%)
	Others	3 (6.4%)
LDH	<200 IU/L	21 (44.7%)
	≥200 IU/L	26 (55.3%)
Brain metastasis	Yes	18 (38.3%)
	No	29 (61.7%)
TNM classification (T)	T1-2	22 (46.8%)
	T3-4	25 (53.2%)
TNM classification (N)	N0-1	22 (46.8%)
	N2-3	25 (53.2%)
TNM classification (M)	M0-1a	14 (29.8%)
	M1b-1c	33 (70.2%)
Disease stage	IIIA-C/IVA	20 (42.6%)
	IVB	27 (57.4%)
EGFR-TKI	Gefitinib	13 (27.7%)
	Erlotinib	8 (17.0%)
	Afatinib	15 (31.9%)
	Osimertinib	11 (23.4%)
PD-L1	Positive	29 (61.7%)
	Negative	18 (38.3%)

EGFR, epidermal growth factor receptor; LDH, lactate dehydrogenase; PS, performance status; PD-L1, programmed death ligand-1; TKI, tyrosine kinase inhibitor.

**Table 2 diagnostics-10-01006-t002:** Associations of the patient characteristics with the PFS (log-rank test).

		PFS	*p*
Age (yr)	<70	9.7 (5.9–13.8)	0.564
	≥70	12.8 (7.4–13.2)	
Sex	Male	9.7 (5.9–15.4)	0.713
	Female	11.2 (5.6–12.9)	
Smoking history	Yes	12.3 (7.4–15.9)	0.681
	No	10.7 (5.9–12.9)	
PS	0–1	11.2 (7.7–12.9)	0.561
	≥2	10.7 (1.5–13.8)	
Histology	Adenocarcinoma	11.2 (7.9–12.9)	0.436
	Others	9.1 (2.0–13.8)	
EGFR mutation	Exon 19 Deletion	11.5 (9.1–15.9)	0.308
	Exon 21 L858R	10.7 (5.9–13.2)	
	Others	7.7 (2.5–13.8)	
LDH	<200 IU/L	11.5 (7.9–13.8)	0.320
	≥200 IU/L	9.1 (5.1–12.9)	
Brain metastasis	Yes	8.8 (5.9–12.9)	0.329
	No	12.3 (9.0–14.1)	
TNM classification (T)	T1-2	12.8 (7.7–15.4)	0.621
	T3-4	10.5 (5.6–12.9)	
TNM classification (N)	N0-1	11.5 (8.6–15.9)	0.132
	N2-3	10.7 (5.1–12.9)	
TNM classification (M)	M0-1a	12.3 (2.5–14.1)	0.304
	M1b-1c	9.7 (7.4–12.9)	
Disease stage	IIIA-C/IVA	12.8 (9.0–14.1)	0.245
	IVB	9.1 (5.9–12.9)	
EGFR-TKI	Gefitinib	12.8 (4.5–15.9)	
	Erlotinib	8.8 (2.5–14.1)	0.608
	Afatinib	11.5 (4.6–13.8)	
	Osimertinib	Not reached	
PD-L1	Positive	9.0 (5.1–12.3)	0.029
	Negative	12.9 (9.7–15.4)	

EGFR, epidermal growth factor receptor; LDH, lactate dehydrogenase; PS, performance status; PD-L1, programmed death ligand-1; TKI, tyrosine kinase inhibitor.

**Table 3 diagnostics-10-01006-t003:** Associations of the patient characteristics with the PFS (Cox proportional hazards model).

		HR (95% CI)	*p*
EGFR mutation	Exon 19 deletion	0.49 (0.21–1.16)	0.106
	Exon 21 L858R	1	
	Others	1.27 (0.30–5.41)	0.749
PS	0–1	1.15 (0.44–3.03)	0.775
	≥2	1	
LDH	<200 IU/L	0.85 (0.37–1.97)	0.710
	≥200 IU/L	1	
Disease stage	IIIA-C/IVA	0.50 (0.21–1.16)	0.105
	IVB	1	
PD-L1	Positive	2.84 (1.20–6.72)	0.018
	Negative	1	

EGFR, epidermal growth factor receptor; LDH, lactate dehydrogenase; PS, performance status; PD-L1, programmed death ligand-1.

**Table 4 diagnostics-10-01006-t004:** Associations of the patient characteristics with the status of acquisition of the T790M mutation of EGFR.

		Whole	T790M-Positive	T790M-Negative	*p*
		27	12 (44.4%)	15 (55.6%)	
Age (yr)	<70	18	7 (38.9%)	11 (61.1%)	0.448
	≥70	9	5 (55.6%)	4 (44.4%)	
Sex	Male	16	6 (37.5%)	10 (62.5%)	0.452
	Female	11	6 (54.6%)	5 (45.5%)	
Smoking history	Yes	13	6 (46.2%)	7 (53.9%)	1.000
	No	14	6 (42.9%)	8 (57.1%)	
Histology	Adenocarcinoma	24	11 (45.8%)	13 (54.2%)	1.000
	Others	3	1 (33.3%)	2 (66.7%)	
PS	0–1	18	7 (38.9%)	11 (61.1%)	0.412
	≥2	9	5 (55.6%)	4 (44.4%)	
EGFR mutation	Exon 19 Deletion	9	6 (66.7%)	3 (33.3%)	0.217
	Exon 21 L858R	17	6 (35.3%)	11 (64.7%)	
	Others	1	0 (0%)	1 (100%)	
LDH	<200 IU/L	14	8 (57.1%)	6 (42.9%)	0.252
	≥200 IU/L	13	4 (30.8%)	9 (69.2%)	
Brain metastasis	Yes	13	6 (46.2%)	7 (53.9%)	1.000
	No	14	6 (42.9%)	8 (57.1%)	
EGFR-TKI	Gefitinib	9	6 (66.7%)	3 (33.3%)	0.243
	Erlotinib	7	3 (42.9%)	4 (57.1%)	
	Afatinib	11	3 (27.3%)	8 (72.3%)	
PD-L1	Positive	16	4 (25.0%)	12 (75.0%)	0.022
	Negative	11	8 (72.7%)	3 (27.3%)	

EGFR, epidermal growth factor receptor; LDH, lactate dehydrogenase; PS, performance status; PD-L1, programmed death ligand-1; TKI, tyrosine kinase inhibitor.

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
