# Peer review of "Association of Tumor PD-L1 Expression with the T790M Mutation and Progression-Free Survival in Patients with EGFR-Mutant Non-Small Cell Lung Cancer Receiving EGFR-TKI Therapy"

_diagnostics, 2020, doi:10.3390/diagnostics10121006_

Round 1
Reviewer 1 Report
The manuscript entitled:" Association of tumor PD-L1 expression with the T790M mutation and progression-free survival in patients with EGFR-mutant non-small cell lung
cancer receiving EGFR-TKI therapy" focused on the clinical evaluation of PD-L1 expression in NSCLC resistant patients after first line TKI treatment requires several major changes to be suitable for publication.
- In the abstract section, the authors report the following sentence:" T790M gene mutation was evaluated by Cobas 20 EGFR assay using tissues or humoral specimens" According to this point, the authors should better explain the sample type of enrolled patients. This aspect is fundamental to clarify material and method section. In addition, they should also change "EGFR mutations" in "sensitive EGFR mutations".
- In the introduction section, the authors report the following sentence:" One of the mechanisms of resistance is acquisition of the T790M mutation of EGFR". In my opinion the authors should modify it because the reported acquired resistance mutations is the most common and more frequent among them.
- In the introduction section, the authors should improve this section by describing the actual must test gene panel actually remarkable to administrate NSCLC patients. In my opinion, this aspect is crucial to evaluate the clinical role of these biomarkers.
- In the text, the authors also report the following sentence:" Patients who had been diagnosed as having EGFR-mutant NSCLC and received first-line treatment with EGFR-TKIs were included in this study". According to this aspect a major revision is required. In my opinion the authors should clarify criteria selection for target population enrolled in the study. Among them, they should emphasize on the TKIs used to treat these patients.
In the material and methods section, several revision should be implemented. Please, could the authors defin the minimum percentage neoplastic cell value identified to consider samples adequate for molecular analysis? Then, could they better express adopted cathegories for PD-L1 expression value? Finally, could they describe limit of detection adopted to evaluate molecular results?
- In the results section, please, could the authors analyze PD-L1 score in relation to several clinical cathegories generally identified in clinical practice?
- In the results section, the authors did not mention EGFR exon 18 sensitive mutation. In my opinion, the authors should investigate their role in a dedicated section.
Author Response
We appreciate for your useful comments that have helped us improve our paper. As indicated in the response that follow, we have addressed comments and revised the article.
Comments and Suggestions for Authors
The manuscript entitled:" Association of tumor PD-L1 expression with the T790M mutation and progression-free survival in patients with EGFR-mutant non-small cell lung cancer receiving EGFR-TKI therapy" focused on the clinical evaluation of PD-L1 expression in NSCLC resistant patients after first line TKI treatment requires several major changes to be suitable for publication.
- In the abstract section, the authors report the following sentence:" T790M gene mutation was evaluated by Cobas 20 EGFR assay using tissues or humoral specimens" According to this point, the authors should better explain the sample type of enrolled patients. This aspect is fundamental to clarify material and method section. In addition, they should also change "EGFR mutations" in "sensitive EGFR mutations".
Response: Thank you for your comment. I added explanation in method section as below; 15 patients: tissue specimens, 9 patients: blood samples, 2 patients: cerebrospinal fluid, and 1patient: plural effusion (page 2, line 86-87). And based on your suggestion, I use the term of “sensitive EGFR mutations” in this article.
- In the introduction section, the authors report the following sentence:" One of the mechanisms of resistance is acquisition of the T790M mutation of EGFR". In my opinion the authors should modify it because the reported acquired resistance mutations is the most common and more frequent among them.
Response: I revised the sentence as follows; Most common mechanism of resistance is acquisition of the T790M mutation of EGFR (page 1, line 37).
- In the introduction section, the authors should improve this section by describing the actual must test gene panel actually remarkable to administrate NSCLC patients. In my opinion, this aspect is crucial to evaluate the clinical role of these biomarkers.
Response: Thank you for your comment and I agree with that Next Generation Sequencing (NGS) testing is becoming an important part of cancer clinical practice.
In introduction section, we described about the improvement of prognosis in patients with EGFR mutant NSCLC by EGFR-TKI, acquisition of resistance by T790M mutation, efficacy of osimertinib for these patients, and clinical parameters that associated with detection likelihood of T790M mutation. These reports are based on PCR testing for EGFR mutation. Thus, I regret to say that I feel difficulty to insert a description about NGS testing in introduction section.
- In the text, the authors also report the following sentence:" Patients who had been diagnosed as having EGFR-mutant NSCLC and received first-line treatment with EGFR-TKIs were included in this study". According to this aspect a major revision is required. In my opinion the authors should clarify criteria selection for target population enrolled in the study. Among them, they should emphasize on the TKIs used to treat these patients.
Response: Thank you for your comment. Based on your suggestion, I revised as follows; The inclusion criteria were as follows; 1) patients with cytologically or histologically confirmed NSCLC harboring sensitive EGFR mutations; 2) patients treated with EGFR-TKIs including gefitinib, erlotinib, or osimertinib as first line treatment between 2014 and 2019. The exclusion criterion was as follows; 1) patients whose records lacked data on the tumor PD-L1 status (page 2, line 59-63).
- In the material and methods section, several revision should be implemented. Please, could the authors define the minimum percentage neoplastic cell value identified to consider samples adequate for molecular analysis? Then, could they better express adopted cathegories for PD-L1 expression value? Finally, could they describe limit of detection adopted to evaluate molecular results?
Response: The minimum detection sensitivity for T790M mutation is about 0.1% when using a plasma sample. I added above description in methods section (page 2, line 87-89). A cut off value of TPS of tumor PD-L1 expression adopted in previous clinical trial was 1% or 50%. Thus, I think the definition of 1% or more as positive is considered valid.
- In the results section, please, could the authors analyze PD-L1 score in relation to several clinical cathegories generally identified in clinical practice?
Response: I analyzed the association based on your comment and found that PD-L1 status (positive vs negative) was not associated with any clinical parameters. I added description in result section as bellows (page 5, line 135-139).
PD-L1 status (positive vs negative) was not associated with clinical parameters including age (<70 vs ≥70, p = 1.000), sex (male vs female, p = 1.000), smoking history (yes vs no, p = 0.211), histology (adenocarcinoma vs others, p = 0.276), EGFR mutation (Exon 19 del vs Exon 21 L858R vs others、p = 0.789), PS (0-1 vs ≥2, p = 0.716), LDH (<200 IU/L vs ≥200 IU/L、p = 0.130), brain metastasis (yes vs no, p = 0.759), stage (IIIA-C/IVA vs IVB、p = 1.000) (Fisher’s exact test)
- In the results section, the authors did not mention EGFR exon 18 sensitive mutation. In my opinion, the authors should investigate their role in a dedicated section.
Response: In the present study 3 patients with uncommon mutation were enrolled, including 2 patients with exon 18 sensitive mutation and 1 patient with exon 21 L861Q. I added description in result section as below (page 5, line 140-145).
In the present study 3 patients with uncommon mutation were enrolled in the analysis, including 2 patients with exon 18 G719A and 1 patient with exon 21 L861Q. Log-rank test revealed that median PFS after the initiation of treatment with EGFR-TKI was short as 7.7 months compared to patients with common sensitive EGFR mutation (p = 0.308, log-rank test). However, Cox regression hazard model suggested that there is an association between tumor PD-L1 expression and PFS after the initiation of treatment with EGFR-TKI independently with EGFR mutation status.
Reviewer 2 Report
The authors investigated how progression-free survival (PFS) and proportion of acquisition of EGR T790M mutation after first-line EGFR TKI treatment correlates with PD-L1 TPS in non-small cell lung cancer. Their results suggest that PFS correlates with PD-L1 22C3 expression, defined as TPS equal or higher than 1%. Similarly, their findings suggest that the proportion of acquisition of T790M mutation is higher in the PD-L1-negative (TPS of less than 1%) patient group. Comments to the authors are as follows:
- Please elaborate on the scoring criteria and possibly give a reference.
- Please provide vendor information for cobas and briefly explain the type of test.
- Row 108: Please write deletion instead of del.
- Row 174: Perhaps the authors meant to say common yet different.
Author Response
The authors investigated how progression-free survival (PFS) and proportion of acquisition of EGFR T790M mutation after first-line EGFR TKI treatment correlates with PD-L1 TPS in non-small cell lung cancer. Their results suggest that PFS correlates with PD-L1 22C3 expression, defined as TPS equal or higher than 1%. Similarly, their findings suggest that the proportion of acquisition of T790M mutation is higher in the PD-L1-negative (TPS of less than 1%) patient group. Comments to the authors are as follows:
- Please elaborate on the scoring criteria and possibly give a reference.
- Please provide vendor information for cobas and briefly explain the type of test.
- Row 108: Please write deletion instead of del.
- Row 174: Perhaps the authors meant to say common yet different.
Response: Thank you for your comments. I added the explanation about Cobas EGFR assay and vendor information in method section as follows (page 2, line 84-85); “The presence of T790M was evaluated by the Cobas EGFR real time PCR assay (LSI Medience, Tokyo, Japan)”.
And I revised our paper as follow.
- Exon 19 del → Exon 19 deletion (page 3, line 119).
- common yet another antibody → common yet different antibody (page 9, line 197).
Reviewer 3 Report
In this study, the PD-L1 expression levels and T790M EGFR gene mutation status were examined in tumor specimens obtained from 27 patients with NSCLC. It was concluded that Patients with negative tumor PD-L1 expression showed a longer PFS and a higher proportion of acquisition of T790M mutation of EGFR after first-line treatment with an EGFR-TKI.
As the authors mentioned in Line 192, the major limitation of this study was that it was conducted in a small number of patients. Given such a small sample size, any factor that is different between the two study groups would show up as biomarkers. The authors should consider to include the information on tumor stage, tumor status, lymph node status and distal metastasis status and see if any incomparable tumor stage distributions between groups would potentially bias the results.
Author Response
Comments and Suggestions for Authors
In this study, the PD-L1 expression levels and T790M EGFR gene mutation status were examined in tumor specimens obtained from 27 patients with NSCLC. It was concluded that Patients with negative tumor PD-L1 expression showed a longer PFS and a higher proportion of acquisition of T790M mutation of EGFR after first-line treatment with an EGFR-TKI.
- As the authors mentioned in Line 192, the major limitation of this study was that it was conducted in a small number of patients. Given such a small sample size, any factor that is different between the two study groups would show up as biomarkers. The authors should consider to include the information on tumor stage, tumor status, lymph node status and distal metastasis status and see if any incomparable tumor stage distributions between groups would potentially bias the results.
Response: Thank you for your constructive comment. I present the information on tumor stage, tumor status, lymph node status, distal metastasis status, and disease stage (table 1). And I analyzed the association with PFS by log-rank test (table 2) and Cox proportional hazard model (table3). Cox proportional hazard model including disease stage as independent variable instead of brain metastasis revealed that PD-L1 status was significantly associated with PFS after the initiation of the treatment with EGFR-TKIs. In addition, PD-L1 status (positive vs negative) was not associated with stage (IIIA-C/IVA vs IVB, p = 1.000, Fisher’s exact test) (page 5, line 135-139).
Round 2
Reviewer 1 Report
The manuscript may be accepted in the present form.
Reviewer 3 Report
The authors have addressed all my concerns.